# Learning to Control Free-Form Soft Swimmers

**Changyu Hu**[1]    **Yanke Qu**[1]    **Qiuan Yang**[2,1]    **Xiaoyu Xiong**[2,1]
**Kui Wu**[3]    **Wei Li**[3,4]    **Tao Du**[1,2]
[1] Tsinghua University [2] Shanghai Qi Zhi Institute
[3] LIGHTSPEED   [4] Shanghai Jiao Tong University

## Abstract

Swimming in nature achieves remarkable performance through diverse morphological adaptations and intricate solid-fluid interaction, yet exploring this capability in artificial soft swimmers remains challenging due to the high-dimensional control complexity and the computational cost of resolving hydrodynamic details. Traditional approaches often rely on morphology-dependent heuristics and simplified fluid models, which constrain exploration and preclude advanced strategies like vortex exploitation. To address this, we propose an automated framework that combines a unified, reduced-mode control space with a high-fidelity GPU-accelerated simulator. Our control space naturally captures deformation patterns for diverse morphologies, minimizing manual design, while our simulator efficiently resolves the crucial fluid-structure interactions required for learning. We evaluate our method on a wide range of morphologies, from bio-inspired to unconventional. From this general framework, high-performance swimming patterns emerge that qualitatively reproduce canonical gaits observed in nature without requiring domain-specific priors, where state-of-the-art baselines often fail, particularly on complex topologies like a torus. Our work lays a foundation for future opportunities in automated co-design of soft robots in complex hydrodynamic environments. The code is available at `https://github.com/changyu-hu/FreeFlow`.

## 1 Introduction

Underwater swimming exemplifies nature's ability to generate versatile movement strategies through free-form morphological adaptations—from the traveling waves of eel-like swimmers to the jet propulsion of cephalopods (Dickinson et al., 2000; Hinch et al., 2012). This diversity in soft-body organisms demonstrates how complex yet efficient control emerges from the interplay between body deformations and fluid dynamics, offering inspiration for bioinspired robotics and adaptive underwater systems. However, exploring such capabilities in artificial free-form soft swimmers poses two challenges. First, unlike articulated rigid-body robots with standardized joint-torque actuation, soft bodies require high-dimensional control policies to coordinate continuum deformations across arbitrary morphologies, lacking a unified control paradigm. Second, learning these policies demands physically-grounded simulations that balance computational efficiency with hydrodynamic fidelity—a tradeoff often skewed toward speed in existing frameworks. As a result, current approaches typically resort to morphology-dependent heuristics, e.g., predefined muscle layouts (Min et al., 2019; Ma et al., 2021) or voxel-aligned contractions (Bhatia et al., 2021), which require fine tuning and restrict exploration of control space. Furthermore, although recent simulation environments (Wang et al., 2023a; Xian et al., 2023) enable data-driven control through simplified fluid models, these approximations omit critical hydrodynamic phenomena like vortex shedding—limiting the discovery of efficient gaits observed in biological swimmers.

To address the limitations of domain-expert, morphology-dependent actuation design, we present a unified control framework that automates both deformation space construction and policy learning

for free-form soft swimmers. Our approach is grounded in a key biological insight (Zhang et al., 2022): natural swimmers exploit spatially low-frequency deformation modes to interact efficiently with fluids, rather than activating the infinitely many degrees of freedom in their soft body. Inspired by this, we introduce morphology-agnostic reduced modes, which compactly encode dominant deformation patterns across arbitrary morphologies with a few parameters. We first leverage geodesic farthest-point sampling to distribute control points adaptively over the swimmer's body. Coupled with linear blend skinning (LBS), these points define a deformation basis that interpolates coarse motions across the entire body . We propose a dynamics correction process to further adjust deformations to physically plausible configurations while preserving kinematic intent. This approach also ensures motions are entirely driven by internal forces, avoiding unphysical momentum injection.

To address the accuracy-efficiency tradeoff of existing simulation environments, we develop a GPU-accelerated simulator tailored for learning swimming strategy, ensuring both hydrodynamic fidelity and computational efficiency required for reinforcement learning (RL). Our simulator integrates the Lattice Boltzmann Method (HOME-LBM by Li et al., 2023) for fluid dynamics, for its inherent parallelism and physical plausibility. The soft swimmers are modeled as finite elements in order to express different morphologies freely, integrated with the state-of-the-art GPU solver (Chen et al., 2024). We incorporate a two-way coupling framework, ensuring that body deformations dynamically interact with fluid—a mechanism essential for thrust generation. Our simulator supports training policies on a $128 \times 128 \times 512$ grid in only a few hours, successfully reproducing physically plausible swimming phenomena.

We evaluate our framework on a diverse set of 3D soft swimmer morphologies, from bio-inspired fish to unconventional morphologies (Fig. 2), demonstrating universal applicability. Our method achieves observable movement patterns in the majority of tested models in forward swimming task, achieving a 50% higher success rate in learning effective swimming gaits compared to state-of-the-art baselines (Wang et al., 2023a), which struggle to produce meaningful motion for the majority of tested morphologies. Furthermore, it learns sophisticated behaviors like vortex exploitation, which simplified fluid models cannot capture. From this general framework, gaits corresponding to canonical biological swimming strategies (e.g., undulation, oscillation, pulsation) emerge automatically without prior kinematic assumptions, establishing a robust pipeline for automated soft swimmer control.

In summary, our work presents the following contributions:

1. We introduce a unified, reduced mode control framework for free-form soft swimmers that automates policy learning for various morphologies. Our approach significantly reduces reliance on human-designed priors while preserving deformation expressiveness.

2. We present a GPU-accelerated simulator optimized for learning soft-body swimming strategies, enabling efficient RL training while capturing hydrodynamic phenomena critical to swimming.

3. Our experiments demonstrate state-of-the-art swimming performance over prior works across a diverse set of morphologies.

## 2 Related Work

**Unified control for diverse morphologies** Many studies on the robot design aim at optimizing general control model for different input structures. Some works are based on articulated rigid bodies due to their simplicity, including Zhao et al. (2020); Gupta et al. (2022); Lu et al. (2025). However, the restriction of degrees of freedom (DoFs) of rigid bodies inhibits them from encoding high-DoF motions. On the other hand, many works turn to soft body design instead, including Bhatia et al. (2021) using mass-spring method and Hu et al. (2019); Wang et al. (2023a,b); Spielberg et al. (2019) using material point method (MPM). Besides, there are other soft-body works based on reduced modes (Zhang et al., 2017; Liang et al., 2023; Barbic and James, 2005) and finite element method (FEM) (Ma et al., 2021; Du et al., 2021; Geilinger et al., 2019; Tan et al., 2012). Leveraging the advantage of efficiency in rigid bodies and flexibility in soft bodies, some studies (Liu et al., 2022; Xu et al., 2021; Xu, 2019; Wang et al., 2019; Li et al., 2024) combine these two representations to form a bone-flesh structure. However, these works focus on actuating soft bodies by rigid link, limiting their control to the underlying rigid joints. The task of optimizing a unified control for soft robot with free structure is still worth exploring. Moreover, most work in this field focuses on terrestrial

robot locomotion, and their extension to swimmers is unobvious and non-trivial due to the high computational costs in fluids simulation and the coupling between fluids and solids.

**Robot-learning environments**   There has recently been an increasing interest in and demand for physics-based learning environments in artificial intelligence and robotics research. Most works (Makoviychuk et al., 2021; Xiang et al., 2020; Todorov et al., 2012; Coumans, 2015; Graule et al., 2022; Huang et al., 2021) focus on high-performance learning environments for simulating and controlling rigid or soft robots alone. Among these works, efforts in building learning environments for fluids are less common due to the computational cost of solving physics-based fluids and solid-fluid interactions. Most existing works (Min et al., 2019; Ma et al., 2018; Ren et al., 2022) build elastic swimmers with biomimetic actuators in simplified fluids and learns their swimming skills with deep RL (Min et al., 2019) or differentiable simulation (Ma et al., 2021). As these simplified fluid models overlook fluid properties (e.g., vorticity and incompressibility) and two-way elastic-fluid coupling, learning advanced swimming skills like jellyfish pulsation and handling multiple swimmers are intrinsically difficult (Min et al., 2019) in these works. There are some works involving the simulation of flow field, including Liu et al. (2022); Wang et al. (2023a); Ma et al. (2021); Holl and Thuerey (2024); Xian et al. (2023), but they either do not support fluid-elastic coupling or suffer from sticking artifacts, which limits their capability of modeling diverse and flexible swimming robots.

**Aquatic animal locomotion**   Animal swimming has long been an intriguing research topic in biology (Dickinson et al., 2000; Hinch et al., 2012) and mechanics (Zhang et al., 2022; Lauder, 2015; Costello et al., 2021). Previous works have identified several distinctive swimming skills commonly shared by aquatic animals which can be divided into three mainstream underwater swimming skills – undulation, oscillation, pulsation – which our pipeline can all automatically discover from their representative swimmers' morphologies.

## 3   Swimmer Modeling

Following the body-brain paradigm (Lipson and Pollack, 2000), we model swimmers through two synergistic components: shape (morphology representation) and controller (deformation policy).

### 3.1   Shape Modeling

We represent the geometry of free-form soft swimmers using a volumetric mesh $\mathcal{M} := \{\mathbf{X}, \mathbf{E}\}$ defined by its rest-shape vertices $\mathbf{X} \in \mathbb{R}^{d \times n}$ and its volumetric element structure $\mathbf{E}$, where $d \in \{2, 3\}$ is the dimension of space and $n$ the number of vertices. We also define the deformed vertices as $\mathbf{x}(t) \in \mathbb{R}^{d \times n}$ and the nodal displacements as $\mathbf{u}(t) = \mathbf{x}(t) - \mathbf{X}$, where $t$ denotes the time. This formulation generalizes across 2D and 3D. While our main results focus on 3D tetrahedral meshes, more 2D results are included in the supplementary materials.

### 3.2   Controller Modeling

Soft body control can generally be categorized into external and internal approaches. External approaches apply forces directly to the body. While simple to implement, they often violate momentum conservation and tend to drag the body toward the target rather than generating propulsion through fluid interaction. Internal approaches generate forces by specifying muscle fibers within the soft body, preserving momentum and offering more physically realistic behavior. However, existing methods typically define these muscle fibers either manually by domain experts or through morphology-dependent heuristics. As a result, they suffer from limited control expressiveness, low automation, and insufficient generalization across diverse body designs. We propose a novel internal controller that automatically modulates the entire rest shape $\mathbf{X}$ of the soft swimmer, ensuring momentum conservation and morphology-agnostic control.

**Kinematic displacement field**   In order to obtain efficient control across varying mesh resolutions and diverse morphologies, we adopt reduced modes defined by linear blend skinning (LBS, Jacobson et al., 2014) as a compact, low-dimensional control space. We modify the rest shape by kinematically proposed displacements $\mathbf{u}_{\text{kin}}$ constructed through reduced modes derived from geodesic control points.

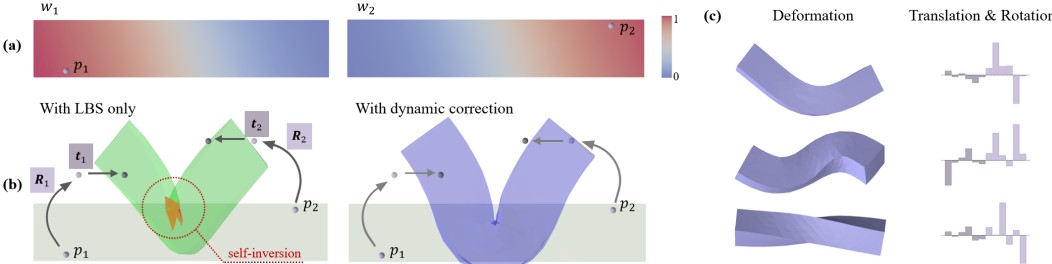

Figure 1: We use a deformable square bar to illustrate our reduced mode control space. Two LBS control points, $p_1$ and $p_2$, are leveraged to generate motions in this example. (a) Upper left shows the normalized weights distributed on the bar, and $w_1$ and $w_2$ are the weights of $p_1$ and $p_2$ respectively. (b) The deformed green mesh (left) is generated by applying vertex-wise weighted combinations of rotations $\mathbf{R}$ and translations $\mathbf{t}$ from control points $p_1$ and $p_2$, with weights $w_1$ and $w_2$. The red region indicates the self-inverted elements. Right figure shows the mesh with dynamic correction. (c) Deformation patterns generated by different distributions of control parameters.

We sample $m$ control points $\mathbf{p}_i$ via farthest-point sampling on rest-shape vertices $\mathbf{X}$ with geodesic distance, which takes into account the mesh's topology and reflects the shortest path in the volume of the mesh. The displacement for each vertex $\mathbf{X}_j$ is calculated as a weighted sum of transformations from all $m$ control points:

$$(\mathbf{u}_{\text{kin}})_j = \sum_{i=1}^{m} w_{ij}(\mathbf{X}_j, \mathbf{p}_i)(\mathbf{R}_i \mathbf{X}_j + \mathbf{t}_i - \mathbf{X}_j) \tag{1}$$

where $\mathbf{R}_i$ and $\mathbf{t}_i$ are learnable rotation and translation modes defined on $\mathbf{p}_i$ and weight $w_{ij}$ determines the influence of control point $i$ on vertex $j$. This function is defined as a radial basis function (RBF) based on the geodesic distance between the vertex $\mathbf{X}_j$ and the control point $\mathbf{p}_i$ (Fig. 1, a). This formulation assigns higher weights to vertices closer to $\mathbf{p}_i$, producing a smooth and spatially localized blend of transformations (see ablation studies in supplementary for details). The weights are normalized on each vertex. This formulation enables resolution-independent control of free-form deformations with only $6m$ degrees of freedom defined on $\mathbf{p}_i$ ($3m$ for rotation and $3m$ for translation).

**Dynamic correction**    While $\mathbf{u}_k$ provides expressive shape changes, it may introduce inverted elements as the LBS formulation ignores the mesh's volumetric integrity (Fig. 1, b left). Inspired by complementary dynamics (Zhang et al., 2020), we compute a correction $\mathbf{u}_d^*$ by solving a perturbation $\mathbf{u}_d$ from the following energy minimization problem:

$$\mathbf{u}_d^* = \arg\min_{\mathbf{u}_d} \Psi(\mathbf{X} + \mathbf{u}_{\text{kin}} + \mathbf{u}_d, \mathbf{X}) + \frac{1}{2} k \|\mathbf{u}_d\|_2^2 \tag{2}$$

where $\Psi(\mathbf{x}, \mathbf{X})$ is the hyperelastic potential energy of a body in its deformed configuration $\mathbf{x}$ relative to its rest configuration $\mathbf{X}$ (see Sec. 4.1), and $k$ a stiffness coefficient that determines the extent of preserving the original deformation modes. This correction projects the kinematically proposed displacement $\mathbf{u}_k$ onto the manifold of dynamically feasible configurations (Fig. 1, b right), while retaining most of the kinematic deformation modes. The total rest shape deformation $\mathbf{u} = \mathbf{u}_{\text{kin}} + \mathbf{u}_d^*$ consists of both the kinematically proposed displacements and dynamic correction.

**Control space properties**    Compared with previous methods that offer only limited or non-physically plausible actuation, the combined displacement field $\mathbf{u}$ satisfies three critical requirements: (1) *Intrinsic actuation* via rest-shape modulation avoids external momentum injection; (2) *Generality* across arbitrary mesh topologies through geodesic sampling; (3) *Compact dimensionality* with $6m$ parameters ($m \ll n$) achieved through LBS control points enabling efficient RL training. As shown in Fig. 1 c, varying coefficients generates diverse rest-shape changes while maintaining dynamical plausibility.

# 4 Swimming Simulation

Efficient underwater locomotion involves rich, dynamic interaction with the surrounding fluid, manifesting complex flow phenomena such as vortex shedding, wake capture and reverse Kármán vortex streets. Accurately capturing these effects requires a simulation framework that balances physical fidelity and computational efficiency. Prior work often relies on simplified fluid or coupling models, limiting the expressiveness of swimmer dynamics (Min et al., 2019; Ma et al., 2021). We address this limitation by integrating state-of-the-art fluid and solid solvers with a GPU-accelerated, two-way coupling scheme.

## 4.1 Elastic Simulation

Given discrete volumetric mesh representation of the soft body, deformed nodal positions $\mathbf{x}$ is governed by the Cauchy momentum equation. The internal elastic forces are derived from the strain energy potential, which depends on the current shape $\mathbf{x}$ and the modulated rest shape $\mathbf{X} + \mathbf{u}$:

$$\mathbf{M}\ddot{\mathbf{x}} + \nabla\Psi(\mathbf{x}, \mathbf{X} + \mathbf{u}) = \boldsymbol{f}_{\text{ext}}, \tag{3}$$

where $\mathbf{M} \in \mathbb{R}^{dn \times dn}$ is the mass matrix, $\Psi$ the strain energy, and $\boldsymbol{f}_{\text{ext}}$ the total external force. We discretize time with standard implicit Euler integration to calculate the updated $\mathbf{x}'$ from the current position $\mathbf{x}$ and velocity $\mathbf{v}$ over a time step $\Delta t$, by iteratively minimizing the incremental potential at each step (Gast et al., 2015):

$$\min_{\mathbf{x}'} \quad \frac{1}{2\Delta t^2}(\mathbf{x}' - \boldsymbol{y})^\top \boldsymbol{M}(\mathbf{x}' - \boldsymbol{y}) + \Psi(\mathbf{x}', \mathbf{X} + \mathbf{u}), \tag{4}$$

where $\boldsymbol{y}$ is the inertia term $\boldsymbol{y} = \mathbf{x} + \Delta t\boldsymbol{v} + \Delta t^2 \boldsymbol{M}^{-1}\boldsymbol{f}_{\text{ext}}$, a constant computed at the beginning of the time step. We utilizes the state-of-the-art GPU-accelerated solvers dedicated to elastics (Chen et al., 2024) to improve computational efficiency.

## 4.2 Fluid Simulation

We consider the lattice Boltzmann method (LBM) as our fluid simulator because it allows for explicit computation of updates. Fluid dynamics can be evolved by a mesoscopic distribution function $f(\mathbf{v}_f, \mathbf{x}_f, t)$, which describes the probability of finding a particle at position $\mathbf{x}_f$ with velocity $\mathbf{v}_f$ at time $t$. The macroscopic quantities of fluid such as density $\rho$ and velocity $\mathbf{v}$ can be derived from $f$. LBM evolves fluid behavior by tracking distribution functions $f$ at discrete lattice nodes on a Cartesian grid. The time integration proceeds through a collision–streaming scheme:

$$f_i(\mathbf{x} + \mathbf{c}_i\Delta t, t + \Delta t) = f_i(\mathbf{x}, t) + \Omega_i(\mathbf{f}), \tag{5}$$

where $f_i$ is the distribution function for lattice direction $\mathbf{c}_i$ and $\Omega_i$ the collision operator which relaxes the distribution function towards a local thermodynamic equilibrium state. We refer interested readers to Lallemand and Luo (2000) for more details.

The explicit nature of LBM's update rule enables massively parallel computation on Cartesian grids. Each lattice node's state is updated independently, minimizing synchronization overhead and maximizing GPU utilization. Our simulator adopts high-order moment-encoded LBM (Li et al., 2023) which achieve higher computational efficiency using less memory while ensuring the accuracy of fluid details.

## 4.3 Elastic-Fluid Coupling

We adopt a weak two-way coupling strategy that alternately updates the fluid and solid at each time step. Compared with other coupling schemes (e.g. strong coupling), it well balances stability and efficiency in the context of learning soft-body swimming controller. The solid influences the fluid through boundary conditions, while the fluid applies pressure forces back onto the solid, which are numerically estimated over the interface. To address the computational challenges posed by extensive fluid-solid interactions, we further develop a fully parallelized intersection detection method that exploits parallelism across both boundary elements and all lattice directions, resulting in significant performance gains. More details are presented in the supplementary materials.

# 5 Swimming-Skill Learning

Building upon the reduced-mode control space and high-fidelity elastic-fluid coupling, we model the task of acquiring locomotion skills as a reinforcement learning (RL) problem. Our physical simulator serves as the dynamic environment, where the agent must learn deformation policies that exploit hydrodynamic interactions to generate thrust.

**Task modeling and policy training.** We model the task as a Markov decision process (MDP) with a state space $\mathcal{S}$ and an action space $\mathcal{A}$. We adopt the standard multi-layer perceptron (MLP) network controller to map the state of a swimmer to actions applied to its actuators. Our simulator serves as the transition function in this MDP that evolves the current state-action pair $(\boldsymbol{s}, \boldsymbol{a})$ to the new state $\boldsymbol{s}'$ after one simulation frame. Each task contains a reward function $R(\boldsymbol{s}, \boldsymbol{a}, \boldsymbol{s}')$ and aims to maximize its discount accumulation in time (Sutton, 2018)$\sum_{i=0} \gamma^i R_i$, where $R_i = R(\boldsymbol{s}_i, \boldsymbol{a}_i, \boldsymbol{s}_{i+1})$ stands for the reward collected in the $i$-th simulation frame. We train all tasks with the soft actor-critic (SAC) method (Haarnoja et al., 2018), a widely adopted deep reinforcement learning (DRL) method known for its stability, sample efficiency, and ability to handle continuous action spaces effectively.

**Unified state representation.** Designing an effective state representation for soft swimmers poses unique challenges: (1) their high-dimensional deformations preclude exhaustive state encoding; (2) morphological diversity demands topology-agnostic observations to avoid case-by-case engineering. To address these, we design a morphology-robust state space $\boldsymbol{s}$ defined as

$$\boldsymbol{s} = \{\boldsymbol{x}_{\text{local}}, \boldsymbol{v}_{\text{local}}, \boldsymbol{v}_{\text{mean}}, \boldsymbol{d}, l, \boldsymbol{a}_{\text{last}}\}, \tag{6}$$

which includes the local positions $\boldsymbol{x}_{\text{local}}$ and velocities $\boldsymbol{v}_{\text{local}}$ of a set of sample points on the model, the average velocity of all vertices $\boldsymbol{v}_{\text{mean}}$, the direction $\boldsymbol{d}$, distance to the target position $l$ and the action of the last step $\boldsymbol{a}_{\text{last}}$ for a typical smooth term (see Eq. 7). In our implementation, we take the LBS control points as sample points directly. At each step, we treat the current sample points as a point cloud and solve the Procrustes problem (Solomon, 2015) to obtain closest rotation and translation from its original pose. The positions, velocities, and target direction are then transformed into this local coordinate frame to more effectively capture the local deformation patterns of the soft swimmer and ensure the learned policy is rotation- and translation-invariant by construction.

**Action.** Leveraging our novel soft-body control representation, we query an action vector $\boldsymbol{a} \in \mathbb{R}^{6m}$ at each control step, where $m$ is the number of LBS control points (Sec. 3.2). Each control point has 6 degrees of freedom—3 for translation and 3 for rotation—within bounded ranges to ensure plausible motion. The number of control points can be fixed or manually specified, enabling resolution-independent control.

**Reward.** Since we use LBS control points and the weights are normalized on each vertex, the mapping from the action space to the deformation space is not injective but exhibits some redundancy. For instance, when all the control points take the same action of rotation and translation, there is no actual actuation applied to the model because of the unchanged rest shape. Therefore, we employ a penalty term in the reward to restrict the redundant degrees of freedom in action space. A typical smooth term is also added. The reward is defined as

$$\begin{aligned}
R &= R_{\text{task}} + \lambda_{\text{smooth}} p_{\text{smooth}} + \lambda_{\text{reg}} p_{\text{reg}}, \\
R_{\text{task}} &= \boldsymbol{v}_{\text{mean}} \cdot \boldsymbol{d}, \\
p_{\text{smooth}} &= -||\boldsymbol{a} - \boldsymbol{a}_{\text{last}}||_2^2 / (6m), \\
p_{\text{reg}} &= -||\boldsymbol{a}||_2^2 / (6m).
\end{aligned} \tag{7}$$

It consists of three components: a task-specific term $R_{\text{task}}$, a smoothness term $p_{\text{smooth}}$ with coefficient $\lambda_{\text{smooth}}$ that encourages natural actions, and a regularization term $p_{\text{reg}}$ with coefficient $\lambda_{\text{reg}}$ that penalizes redundant actions. The task-specific terms of reward $R_{\text{task}}$ is the dot product of velocity and the direction to target. See supplementary materials for details.

**Task Setup.** Our primary evaluation focuses on the forward swimming task in 3D. To further probe the versatility of our framework, we also introduce three advanced tasks evaluated on a 2D swimmer: target navigation, energy-efficient locomotion, and flow resistance. The detailed setup and results for these tasks are presented in the supplementary materials, demonstrating the framework's adaptability to diverse objectives.

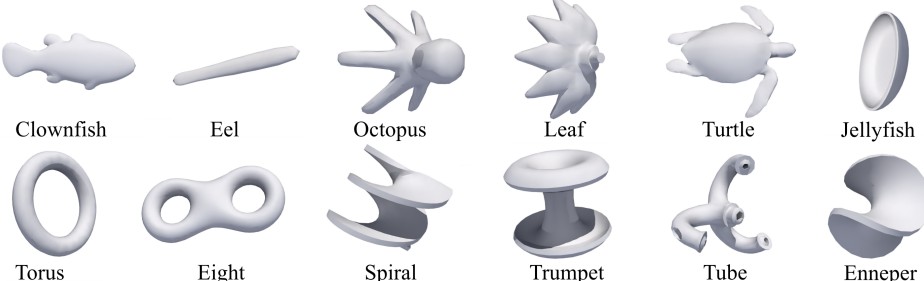

Figure 2: A collection of swimmer morphologies used in our experiments. The top six are bionic morphologies, while the bottom six are abstract morphologies with unconventional topologies.

# 6 Results

## 6.1 Experimental Setup

**Dataset**   We construct a novel collection of 12 soft swimmer morphologies (Fig. 2), including 6 bio-inspired and 6 abstract morphologies. The bio-inspired morphologies cover representative swimming mechanisms implemented by aquatic animals after millions of years of evolution in nature: eel-like undulation, octopus-like oscillation, and jellyfish-like pulsation. The abstract morphologies are deliberately designed to test the framework's ability in unconventional swimming scenarios beyond biological templates—scenarios where effective deformation patterns may not exist. These morphologies intentionally lack obvious deformation pathways for propulsion, forcing the controller to discover novel fluid-structure interaction strategies through exploration. All meshes are normalized to the same scale and tetrahedralized by fTetWild (Hu et al., 2020), comprising about 400 to 1,500 vertices and 1,000 to 6,000 finite elements.

**Baselines**   We evaluate our method against three baselines:

*Domain-expert controller.* Following expert-designed templates (Lin et al., 2019), we implement manually tuned actuators for well-understood morphologies—axial muscles for clownfish and circular muscles for jellyfish. We directly transfer the clownfish's muscle design to the eel since they are geometrically analogous. For the torus, we apply four segments of tangential-direction muscles following the actuation approach outlined in DiffPD (Du et al., 2021) for terrestrial environments.

*Clustering-based controller.* DiffuseBot/SoftZoo (Wang et al., 2023b,a) are two state-of-the-art approaches in soft swimmer control, so we adopt their clustering-based method as one of the SOTA baselines. Following the approach, we segment swimmers into user-defined body regions via K-means clustering on centers of finite elements and then use principal component analysis (PCA) to extract dominant deformation directions for each region to define muscle orientations.

*Differentiable controller.* We test SoftZoo's controller design in their open-sourced pipeline (Wang et al., 2023a). For comprehensive comparison, we adapt the framework by freezing morphology and material properties to isolate actuator optimization effects. However, this baseline faces critical technical limitations in our experimental setting: (1) its differentiable MPM simulation becomes numerically unstable beyond 3 seconds of simulated time (compared with 15 seconds in our task), causing gradient explosions that prevent policy convergence; (2) It fails to achieve $128 \times 128 \times 512$ spatial resolution due to memory constraints. These two factors prevent the baseline from completing all 12 experiments under our temporal and spatial settings. We therefore exclude it from quantitative comparisons. Qualitative results in the supplementary materials show that its learned policies tend to repeat similar stretching patterns with limited diversity.

More details for all baselines are provided in our supplementary materials.

## 6.2 Quantitative Results

As shown in Tbl. 1, our method outperforms baseline controllers across most morphologies in the forward-swimming task. Performances are evaluated by rewards reflecting swimming distance.

Table 1: Normalized reward (mean ± std over 5 trials) for the forward swimming task. Bold indicates the best performance per morphology; gray entries denote controllers failing to make a visible movement (reward less than 0.3).

| Method | Model | | | | | |
|---|---|---|---|---|---|---|
| | Clownfish | Eel | Octopus | Leaf | Turtle | Jellyfish |
| Domain-expert | **11.96** ± 0.10 | 0.17 ± 0.06 | - | - | - | 18.85 ± 0.42 |
| Clustering-based | −1.12 ± 0.22 | **13.2** ± 1.0 | 0.15 ± 0.05 | **1.78** ± 0.07 | −1.18 ± 0.16 | 0.87 ± 0.13 |
| Ours | 10.34 ± 0.42 | 6.26 ± 1.18 | −0.03 ± 0.03 | 0.88 ± 0.29 | **8.67** ± 0.94 | **23.43** ± 1.14 |
| | Torus | Eight | Spiral | Trumpet | Tube | Enneper |
| Domain-expert | −0.31 ± 0.31 | - | - | - | - | - |
| Clustering-based | −0.07 ± 0.01 | −0.04 ± 0.01 | −0.13 ± 0.02 | 0.16 ± 0.05 | 1.02 ± 0.56 | 0.20 ± 0.03 |
| Ours | **15.05** ± 1.27 | **3.99** ± 0.43 | **3.20** ± 0.37 | −0.16 ± 0.22 | **3.99** ± 1.24 | **12.33** ± 1.02 |

Domain-expert designed muscle templates (Fig. 3 bottom two rows) perform well on bio-inspired shapes like the clownfish, achieving up to 115% of our method's performance due to their well-understood swimming patterns. However, this advantage quickly deteriorates on eel (3%), which is geometrically similar but different in proportion of its parts, revealing high sensitivity to geometric changes. Moreover, muscle templates originally optimized for terrestrial locomotion (e.g., torus) exhibit poor transferability underwater (fails), highlighting the limits of human intuition.

The clustering-based controllers implemented by one of the SOTA methods in soft swimmer learning fails to produce effective gaits for 8 out of 12 morphologies and breaks down entirely on abstract shapes, often resulting in unstable oscillations or nearly motionless poses. This is because the clustering method restricts deformations to the principal axes of precomputed muscle fibers, insufficient to produce extensive fluid interaction necessary for effective swimming.

In contrast, our framework demonstrates robust performance, enabling over 80% of the tested morphologies to achieve forward locomotion. This high success rate, particularly on unconventional shapes like the torus and Enneper surface, suggests a broader implication: swimming potential may be a latent property in a wider range of geometries than previously assumed. This generality stems from our automated pipeline, which adapts naturally to diverse topologies, and our novel rest-shape deformation strategy, which enables expressive yet effective motions. Consequently, our framework acts not just as a controller, but as a computational tool to reveal and realize the swimming aptitude of arbitrary designs, challenging prior assumptions about what constitutes a viable swimmer.

**Training Stability**   To address the stochastic nature of reinforcement learning, we evaluated the training stability of our framework. We trained policies for a fish-like swimmer in 2D across six independent runs with different random seeds while keeping all hyperparameters constant. The learning process proved to be highly consistent, with all runs converging to a similar high level of performance. The final mean normalized reward was 6.71 with a low standard deviation of 0.77. The learning curves, which we detail in the supplementary materials, show a stable and monotonic increase in reward, confirming that our method is robust and its performance is reproducible.

## 6.3   Qualitative Analysis

Our method learns effective swimming strategies across diverse morphologies (Fig. 3), producing biologically plausible motions for novel topologies. Several key observations are summarized below. Please refer to our videos in the supplementary materials for their full motions.

**Torus**   The torus-shaped swimmer achieves propulsion through a periodic deformation cycle, dynamically balancing body-fluid momentum exchange (Fig. 3, a. left). First, the torus undergoes controlled self-twisting into an 8-shaped configuration. Then it obtains angular momentum from fluid and starts to rotate, generating vortex-induced forces for forward motion. In contrast, the clustering baseline struggles to make a movement (Fig. 3, a. right).

**Enneper Surface**   Resembling a saddle-shaped skirt, the swimmer performs rhythmic stretching/relaxation cycles, creating a "dancing" motion that leverages pressure gradients across its curved surface (Fig. 3, b. left). This emergent behavior achieves stable locomotion despite the morphology's

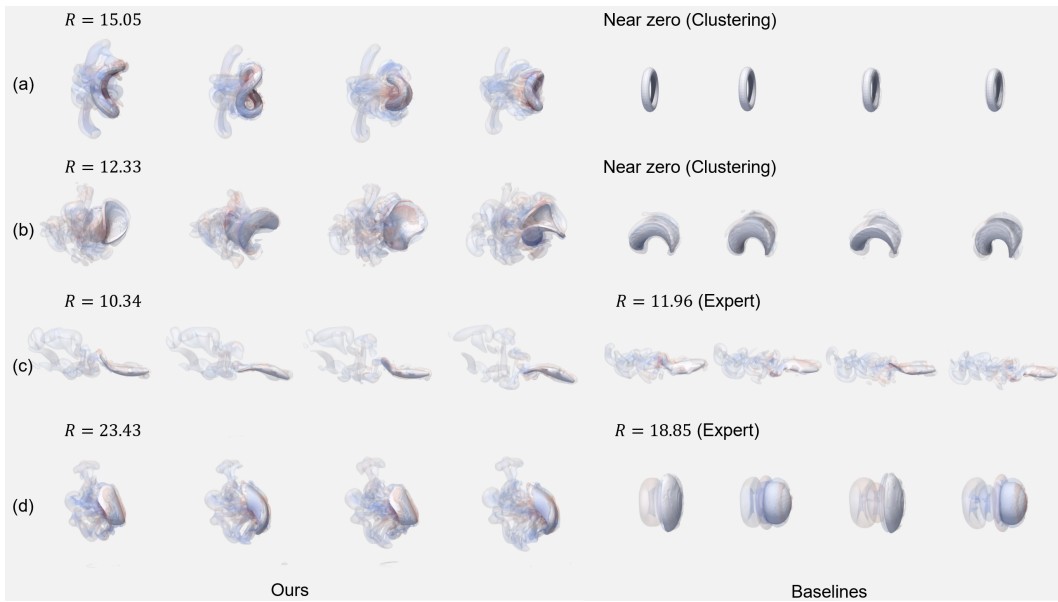

Figure 3: Key frames of some swimmers' motions: (a) torus (b) Enneper surface (c) clownfish (d) jellyfish. Swimmers in (a) and (b) are compared with clustering baseline, while swimmers in (c) and (d) are compared with domain-expert baseline. The structures of fluid field are visualized by extracting the isosurface of $q$ criterion (Hunt et al., 1988) of velocity field.

negative Gaussian curvature. In contrast, the clustering baseline make slight deformation and can hardly move (Fig. 3, b. right).

**Eel/Clownfish** The controllers produce traveling-wave body undulations (Fig. 3, c. left), qualitatively reproducing the canonical undulatory propulsion common to natural anguilliform (eel-like) and carangiform (fish-like) locomotion (Fig. 3, c. right). Leveraging our realistic LBM fluid simulator, the swimmers effectively harness vortex shedding from the tail for propulsion, closely aligning with biological observations—an effect unattainable in simplified simulation environments (Ma et al., 2021; Wang et al., 2023a).

**Jellyfish** For the jellyfish, the learned policy is based on pulsation-based propulsion, a canonical swimming mode. However, instead of the synchronized bell contraction common in nature (Fig. 3, d. right), our policy discovers a novel variant that propels fluid through alternating contractions along two mutually orthogonal directions (Fig. 3, d. left). It is worth noting that this strategy achieves higher speed than the domain-expert actuation design described above (Sec. 6.1).

### 6.4 Extensions to Energetic Efficiency

Beyond maximizing travel distance, a key performance metric for both biological and robotic swimmers is energetic efficiency. To demonstrate that our framework can optimize for such objectives, we conducted an experiment to learn energy-efficient gaits. Following established biomechanics literature Verma et al. (2018), we define energy cost as the total work done by the internal forces to deform the swimmer's body. This physically-grounded metric is efficiently computed at each simulation step.

We augmented the reward function with an energy penalty term: $R_{\text{eff}} = R_{\text{task}} - w_e \cdot E$, where $E$ is the energy cost and $w_e$ is a tunable penalty coefficient. We trained policies for the clownfish morphology with varying $w_e$ and evaluated the trade-off between distance and efficiency using the **Cost of Transport (CoT)**, defined as total energy consumed per meter traveled.

The results in Tbl. 2 show a clear and predictable trade-off. As the energy penalty increases, the learned gaits become more conservative, consuming significantly less energy and achieving a better CoT. An excessively high penalty ($w_e = 0.05$) correctly suppresses movement almost entirely,

Table 2: CoT (lower is better) and travel distance for different energy penalty weights ($w_e$).

| $w_e$ | Forward Distance | Energy Cost | CoT |
|---|---|---|---|
| 0 (Baseline) | **1.68 m** | 433.2 J | 258.3 J/m |
| 0.005 | 1.57 m | 211.0 J | 134.3 J/m |
| 0.02 | 1.06 m | **78.8 J** | **74.7 J/m (Optimal)** |
| 0.05 | -0.01 m | 0.3 J | N/A |

confirming that the policy robustly optimizes the combined objective. This demonstrates the flexibility of our framework to incorporate and optimize for complex, physically-grounded objectives beyond simple locomotion. The precise formulation for the energy cost is detailed in the supplementary material.

## 6.5 Ablation Studies

In this section, we conduct a series of ablation studies to analyze key components of our method, including the effect of control point count on motion, the selection of geodesic distance in LBS process, the choice of LBM for fluid simulation, and momentum conservation enabled by internal actuators. Results show that: (1) geodesic distance proves critical for capturing geometry-aware deformation modes compared to Euler distance; (2) the number of control points may affect the magnitude and the complexity of the motion, depending on the morphology; (3) our LBM fluid solver captures hydrodynamic details for swimming where simplified fluid model fails; (4) we ablate fluid interactions to show that our internal actuator does not introduce non-physical momentum.

Details including figures and videos can be found in our supplementary materials.

## 7 Conclusions

This work presents a unified framework for learning to control free-form soft swimmers. By constructing morphology-agnostic reduced control spaces through LBS and dynamics correction, our framework automates actuator design across diverse morphologies, bypassing the need for domain-expert manual design while preserving physical consistency. Coupled with a GPU-accelerated simulator resolving vortex-mediated thrust mechanisms, our approach enables the emergence of bio-inspired gaits without domain-expert priors and generalizes to unconventional morphologies where prior methods fail, and demonstrates robust performance on a range of advanced locomotion tasks.

While our framework advances automated control, several open challenges remain. Our focus on fixed morphologies paves the way for future work in full morphology-control co-design, for which our unified controller and high-fidelity simulator provide a critical foundation. Another significant hurdle is sim-to-real transfer; the policies learned in our idealized environment can serve as a vital baseline for future research aimed at bridging the reality gap. Finally, while our method learns specialized policies, developing a universal controller that generalizes across unseen soft-body geometries in fluid remains a challenging open problem, likely requiring breakthroughs in meta-learning.

## Acknowledgments and Disclosure of Funding

We would like to thank Dr. Chao Yu for her valuable advice on reinforcement learning training. Tao Du acknowledges the research funding support from Tsinghua University and Shanghai Qi Zhi Institute, and Wei Li benefits from SJTU's startup funds.

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
