# OpenReview forum: "Learning to Control Free-Form Soft Swimmers"
_NeurIPS.cc/2025/Conference — NeurIPS 2025 poster_

### Official Review · Reviewer_SDpb · 2025-06-30

**Clarity:** 3
**Significance:** 3
**Originality:** 3
**Rating:** 5
**Confidence:** 4

**Summary:**

This paper introduces a unified learning framework for controlling free-form soft-bodied swimmers by combining a reduced-mode internal actuation model with a GPU-accelerated fluid-structure simulator based on the Lattice Boltzmann Method (LBM). The system learns swimming policies through reinforcement learning (SAC) in a morphology-agnostic control space, achieving notably strong performance across a wide range of both bio-inspired and abstract 3D morphologies. Notably, the method does not rely on predefined muscle layouts or hand-crafted actuation heuristics, enabling the emergence of plausible swimming behaviors through physically grounded simulation.

**Questions:**

1.	Behavioral Diversity: Can your framework support learning of multiple distinct swimming strategies (e.g., hovering, turning, darting) within a single morphology? Would introducing multi-task learning or a goal-conditioned policy allow this? Demonstrating this capability would significantly strengthen your claims of biological plausibility.
2.	Gait Transitions: Is it possible to transition smoothly between different learned gaits within the same controller or policy space? If not, do you foresee how your reduced-mode control representation might be extended to support this?
3.	Biological Framing: Could you clarify your use of phrases like “rediscovering evolved behaviors”? As it stands, the methodology does not explicitly model evolutionary processes. Framing this as emergent similarity rather than discovery could help avoid misunderstanding.
4.	Co-design Potential: You mention future work on joint morphology-control optimization. Do you believe your current control representation is compatible with a differentiable or evolutionary co-design pipeline?
5.	Additional Tasks: Have you considered extending your experiments to include non-trivial objectives such as turning, avoiding obstacles, or energy-efficient locomotion? Such tasks would better demonstrate policy generalization and control flexibility.

**Ethical Concerns:**

["NO or VERY MINOR ethics concerns only"]

**Final Justification:**

The primary concern I had has been sufficiently clarified in the rebuttal. Accordingly, I have decided to update my evaluation to 'accept

**Limitations:**

While the paper discusses some limitations, such as the absence of joint morphology-control co-design, several important points warrant further consideration. First, the method is restricted to learning a single locomotion strategy per morphology and does not support multi-modal behaviors or context-driven gait transitions—key features of biological locomotion. Second, the task setup is narrowly focused on forward swimming, limiting the evaluation of adaptability or robustness across diverse goals. Third, morphologies are fixed and not co-optimized with control, missing the opportunity to study the interdependence of structure and behavior. Finally, the method is evaluated entirely in simulation without addressing real-world deployment challenges such as actuation delay, sensor noise, or sim-to-real transfer. While the framework is promising, acknowledging these aspects would provide a more balanced view of its current scope and future potential.

**Paper Formatting Concerns:**

None.

**Quality:**

3

**Strengths And Weaknesses:**

Strengths
•	The integration of geodesic-based sampling, linear blend skinning (LBS), and LBM-based fluid simulation forms a well-designed and efficient pipeline for learning physically plausible control policies in fluid environments.
•	The method is applied successfully to a broad set of morphologies, including highly unconventional topologies where baseline controllers struggle, demonstrating its robustness and generalizability.
•	The learned controllers exhibit swimming behaviors reminiscent of biological strategies (e.g., undulation, pulsation) without requiring any explicit kinematic priors, which is an appealing result.
•	The authors provide thorough ablation studies and quantitative comparisons to validate the design choices.

Weaknesses
1.	Limited behavioral diversity and adaptability
One of the defining characteristics of biological swimmers is their ability to exhibit multiple distinct locomotion modes (e.g., hovering, darting, gliding) using the same body structure, and to transition between these modes smoothly in response to environmental changes.
In contrast, the proposed system is trained on a single task objective (forward swimming) and results in a single learned gait per morphology. There is currently no mechanism to support multi-modal behaviors or gait switching, which limits the biological plausibility and adaptive capacity of the framework.
2.	Ambitious biological framing not fully supported by methodology
The paper frequently refers to the "rediscovery" or "understanding" of evolved swimming strategies, but it does not model evolutionary or developmental processes. Morphologies are fixed, and control policies are optimized independently for each shape. While the resulting behaviors are certainly interesting and sometimes biologically plausible, these claims may be overstated given the actual scope of the method. A more modest and precise framing would strengthen the overall presentation.
3.	Limited novelty in methodology
The proposed method combines existing components: reduced-mode actuation via LBS, geodesic sampling, LBM-based simulation, and standard SAC reinforcement learning. These are all well-established techniques. The originality lies more in their practical integration than in introducing fundamentally new algorithms or theoretical insights.
4.	Simplified task setup
All training scenarios focus on a forward swimming task with a fixed target direction and reward function. There is little exploration of more complex objectives such as turning, obstacle avoidance, or energy-efficient maneuvering. This restricts the evaluation of behavioral richness and policy flexibility, and may limit the method’s applicability to real-world adaptive tasks.

---

> ### Author Rebuttal · Authors · 2025-07-31
>
> We sincerely thank the reviewer for the thorough and constructive feedback on our manuscript.
>
> **[L2] Focused on Forward Swimming**
>
> We would like to clarify that our framework indeed supports more tasks than just forward swimming:
>
> - **Navigation Task**: Our supplementary material and video already present this task on both our 3D jellyfish and 2D fish morphologies. This task demonstrates the agent's **adaptability** by requiring it to **dynamically adjust** its swimming gait to pursue a series of on-the-fly generated targets, showcasing its ability to handle **diverse goals** within a single episode.
> - We have also conducted two new experiments on **energy-efficient swimming** and **flow resistance**, which we will detail in our subsequent responses.
>
> In summary, our framework is not limited to forward swimming. These experiments demonstrate its capability to train policies that exhibit adaptability and can achieve diverse goals.
>
> ---
>
> **[W1/Q1/L1] Multiple Learned Gaits on Single Morphology**
>
> We thank the reviewer for this excellent point on behavioral diversity. We believe our **navigation task** already demonstrates the adaptive capability in question. In this task, the learned policy on the 2d fish morphology exhibits **context-driven gait switching**: it naturally adopts a **turning** gait that exhibits **asymmetric** deformation to turn towards a target, then seamlessly transitions to a **darting** gait that exploits **symmetric** deformation for forward propulsion (Figure 5 in supplementary material and fish2d.mp4 in the media attached).
>
> This adaptive behavior emerges from a single, simple reward function, which shows a promising sign that our framework's capacity to learn multiple strategies and seamless switching without explicit programming for each.
>
> Furthermore, to clarify the "single task objective" point, our new **energy-efficient swimming** experiment demonstrates the objective's flexibility. By introducing an energy penalty into the reward, we elicit distinct gaits from the same morphology: one optimized for maximum speed and another for energy efficiency. This shows our framework can produce a spectrum of behaviors by simply adjusting the task objective.
>
> ---
>
> **[Q2] Smooth Transitions between Different Gaits**
>
> As we explained in our response to [W1/Q1/L1] above, our navigation task demonstrates that a single policy can indeed produce smooth, context-driven gait transitions.
>
> When the target deviates from the agent's path, the policy generates an **asymmetric gait**: it produces a stronger body bend towards the target and a weaker counter-stroke, creating the necessary yaw torque to change direction. Once aligned, the policy seamlessly transitions back to a **symmetric, rhythmic gait** to maximize forward propulsion.
>
> Our **reduced-mode control representation** is particularly well-suited for this. It parameterizes the holistic deformation of the body, allowing the policy to learn to smoothly modulate just a few control parameters to create either the large-scale asymmetric bends needed for turning or the balanced oscillations for efficient forward swimming. This demonstrates that our representation is sufficiently expressive to support this form of adaptive control.
>
> ---
>
> **[W4/Q5] Simplified Task Setup / Additional Tasks**
>
> We thank the reviewer for the suggestions, which highlight the importance of demonstrating our framework's versatility on more complex tasks. To this end, our experiments now cover three such tasks:
> - **Navigation**: Our navigation task, already present in the supplement, directly addresses the turning task requested in this question. The agent learns to dynamically adjust its gait to follow the target, demonstrating adaptive turning.
> - **Energy-Efficient Locomotion**: We have conducted new experiments where an energy penalty is added to the reward. The agent successfully learns a distinct, smoother gait that minimizes the energy cost, showcasing its ability to optimize for efficiency.
> - **Flow Resistance**: We have added a new task where the agent must maintain its position against a persistent fluid flow, following DiffAqua [1]. This demonstrates the policy's ability to generate continuous corrective forces to maintain stability.
>
> For a detailed analysis of the energy-efficient locomotion task, we respectfully refer the reviewer to our response to **Reviewer Z1aA [Q2]**, where this is addressed in full.
>
> **Flow Resistance**
>
> We trained a 2D fish policy to maintain its position against a persistent head-on fluid flow. The policy was rewarded for minimizing its distance from the starting point. The learned policy demonstrated good stability.
>
> |Policy|Average Deviation (m)|
> |---|:---:|
> |**Trained**|0.030|
> |**Zero Action**|0.191|
> |**Random**|0.226|
>
> This shows the agent learned to generate persistent, continuous counter-thrust to actively resist the environmental disturbance.
>
> ---
>
> **[W2/Q3] Biological Framing**
>
> Thank you for this valuable feedback. Our biological connection is indeed qualitative: the learned behaviors match canonical swimming modes, such as the **undulation** in our eel/clownfish models and the **pulsation** in our jellyfish model (Fig. 4). Crucially, these distinct modes emerged autonomously from our framework without explicit priors. We agree our use of "rediscovering evolved behaviors" was imprecise and, per your suggestion, will revise our text to focus on the emergent similarity of the behaviors rather than “discovery”.
>
> ---
>
> **[W3] Limited Novelty in Methodology**
>
>  While we employed quite a few established techniques, we believe the merits of our framework also include their novel adaptation and **non-trivial integration** of a system that balances the competing demands of physical fidelity and computational speed required for soft robot learning.
>
> Our methodological novelty lies in combining LBS with complementary dynamics. While LBS is a standard kinematic tool, we explore its application in a physics-based context where it defines a low-dimensional control space, and complementary dynamics ensure the resulting holistic deformations are physically plausible.
>
> Furthermore, the design decisions for assembling a complete system that supports RL for complex swimming behaviors are highly non-trivial. The community has explored several technical combinations, including CFD with A2C [2], differentiable MPM for solids and fluids [3], or simplified fluid models [4]. However, these approaches often present trade-offs that limit their use for the free-form soft swimmer control we demonstrate: full CFD is often too slow for RL, differentiable simulators can struggle with the long-horizon stability required to learn robust controllers, and simplified models lack the physical realism to capture complex hydrodynamics.
>
> Therefore, we believe that identifying and successfully integrating our specific combination of techniques is an original and significant result. It provides the community with a practical and effective platform that balances physical fidelity, computational efficiency, and generality, opening up new possibilities for research in this domain.
>
> ---
>
> **[Q4] Co-design Potential**
>
> We thank the reviewer for this question. We believe our representation is compatible with both co-design paradigms.
>
> **Integration into a Differentiable Pipeline**
>
> The LBS computation is differentiable. By treating the control points as optimizable variables, our LBS-based kinematic mapping becomes a fully differentiable module. Furthermore, the complementary dynamics solver is also differentiable through standard adjoint methods. Therefore, our entire physics-based control module can be integrated into a gradient-based pipeline.
>
> **Integration into a Gradient-free Pipeline**
>
> On the other hand, the representation is highly suitable for **gradient-free methods like evolutionary algorithms**. Its **compact and low-dimensional design** introduces minimal computational overhead. This efficiency is a significant benefit for population-based approaches, which require a large number of simulation rollouts to effectively search the design space.
>
>
> ---
>
> **[L3/L4] Limitations on Co-design and Sim-to-Real Transfer**
>
> Thank you for these valuable comments. We agree that joint co-optimization and sim-to-real transfer are both important and challenging avenues for future research. To provide the more balanced view you suggested, we will expand our conclusion to discuss these topics, framing them as key limitations and exciting directions for our work.
>
> We are truly grateful for your insightful comments and constructive feedback. We hope that the additional details and experimental results we provided have effectively addressed your concerns. Please feel free to let us know if there are any remaining concerns. We look forward to hearing your thoughts on our rebuttal.
>
> ---
>
> References:
>
> [1]. Ma, P., Du, T., Zhang, J. Z., ... & Matusik, W. (2021). Diffaqua: A differentiable computational design pipeline for soft underwater swimmers with shape interpolation. ACM Transactions on Graphics (TOG), 40(4), 1-14.
>
> [2]. Zhang, T., Tian, R., Yang, H., ... & Xie, G. (2022). From simulation to reality: A learning framework for fish-like robots to perform control tasks. IEEE Transactions on Robotics, 38(6), 3861-3878.
>
> [3]. Wang, T. H., Ma, P., Spielberg, A. E., ... & Gan, C. (2023). SoftZoo: A Soft Robot Co-design Benchmark For Locomotion In Diverse Environments. In The Eleventh International Conference on Learning Representations.
>
> [4]. Min, S., Won, J., Lee, S., Park, J., & Lee, J. (2019). Softcon: Simulation and control of soft-bodied animals with biomimetic actuators. ACM Transactions on Graphics (TOG), 38(6), 1-12.

---

> > ### Comment · Reviewer_SDpb · 2025-08-04
> >
> > Thank you very much for your thoughtful and detailed response. The additional experiments have clarified the concern I was most worried about. Based on this, I would like to raise my score and change my evaluation to a Borderline Accept.

---

> > > ### Author Response · Authors · 2025-08-05
> > >
> > > Thank you very much for your reply and positive feedback. We are glad that the additional experiments provided have resolved your primary concern.
> > >
> > > We deeply appreciate your positive re-evaluation and recognition of our efforts. We are happy to provide any further clarification that might be needed during your discussion.

---

> > > > ### Comment · Reviewer_SDpb · 2025-08-06
> > > >
> > > > Thank you very much for your courteous and thoughtful response. I found your paper very interesting and enjoyed reading it. The additional experiments have clarified the concerns I was most worried about, and I believe the results you have presented are already sufficiently strong.
> > > >
> > > > If I may make a small request, it would be very helpful if you could also update the videos to include the additional experiments.
> > > > If the paper is accepted, I would look forward to seeing the final version with the updated materials.

---

> > > > > ### Author Response · Authors · 2025-08-07
> > > > >
> > > > > Thank you for the suggestion! We will certainly update the supplementary videos to include these new results, ensuring the final version of the paper is as comprehensive as possible.
> > > > >
> > > > > Thank you again for your valuable feedback and support!

---

### Official Review · Reviewer_xudV · 2025-07-02

**Clarity:** 3
**Significance:** 3
**Originality:** 3
**Rating:** 5
**Confidence:** 3

**Summary:**

This paper proposes an automated framework featuring a unified reduced-mode control space that naturally captures typical deformation patterns observed across diverse and unconventional swimmers, and a GPU-accelerated simulator to efficiently resolve fluid-structure interactions, specifically tailored for learning soft-body swimming controllers. The proposed method is evaluated on 12 soft-body morphologies and benchmarked against domain-expert designed and state of the art (clustering-based and differentiable) control approaches. The proposed approach performs similarly to the state of the art for the biology based swimming morphologies and outperforms the state of the art for abstract swimmers.

**Questions:**

Currently, each swimmer is trained from scratch, it is unclear whether the trained policy can be generalized or transfer to unseen morphologies. How can this be addressed?
Please clarify the computational cost and training speed of the simulator and training method.
Please provide more details of each reward functions and how reward weights were chosen. Could a poorly tuned reward limit the emergence of complex locomotion strategies?

**Ethical Concerns:**

["NO or VERY MINOR ethics concerns only"]

**Final Justification:**

Good contribution.

**Limitations:**

Yes

**Paper Formatting Concerns:**

No issues

**Quality:**

3

**Strengths And Weaknesses:**

Strengths:
The paper introduces a framework that unities soft-body control with reduced deformation modes and implemented with a GPU-accelerated simulator. The method is evaluated on 12 diverse morphologies, six intuitive biology-inspired swimmers and six abstract shapes. The evaluation experiments are well-designed and sufficient to support the paper's claims.

The paper is well-written and easy to follow. The figures and videos are helpful for understanding this work.

The framework could be used for co-design problems for both morphology and control or for studying locomotion when the analytical modelling is challenging.

The integration of a custom LBM-based GPU simulator to resolve dynamics in RL is novel. Furthermore, the combination of geodesic-based-LBS and dynamic correction for internal actuation is novel as far as I understand.

The evaluation of the approach is rigorous and thorough. The approach outperforms the state of the art for unconventional shapes and performs similarly to the other approaches for the biologically-inspired shapes.

The approach is shown to generalize to a set of arbitrary morphogenesis, something that prior work struggles with.

Weakness:
There is no sim-to-real discussion in this paper or interpretation on why the controllers work.
The paper hints the implementation of automated morphology-control co-design, but it does not implement joint optimization.
The framework does not yet support general controllers that transfer across unseen shapes, each policy is trained per-shape from scratch. Also, the performance of the controllers under different conditions (eg disturbances) would be another interesting evaluation
There is no discussion on the complexity and/or computational cost of the approach, or comparison with state of the art across these lines. This is a big challenge for the field as far as I understand.

---

> ### Author Rebuttal · Authors · 2025-07-31
>
> We thank the Reviewer for the positive evaluation and for providing detailed, constructive feedback on our work. We are encouraged by the reviewer's assessment and have found the suggestions regarding generalization, robustness, computational cost, and reward design to be very helpful for improving our manuscript.
>
> **[W1] Interpretation on Why the Controllers Work**
>
> On a physical level, the controllers work by discovering and executing time-varying body deformations that generate effective fluid-structure interactions for propulsion. Our analysis shows that the reinforcement learning agent, through trial and error, learns to create kinematic patterns that correspond to well-understood principles of thrust generation in fluids:
>
> - For our **undulatory swimmers (eel and fish models)**, the learned policies generate traveling waves along the body that shed vortices in a manner consistent with a **reverse von Kármán vortex street**. This specific wake signature is a known indicator of net thrust production [1].
> - For our **jellyfish model**, the controller learns to produce rhythmic bell contractions that create and shed a **starting vortex ring** with each pulse. This generates an impulsive jet of fluid, propelling the body forward, which is the canonical mechanism [2] for pulsatile swimmers.
>
> Therefore, the interpretation is that our framework enables the agent to learn the fundamental physics of fluid-based locomotion from scratch, successfully identifying gaits that manipulate pressure fields and vortex dynamics to its advantage.
>
> ---
>
> **[W1] Discussion on Sim-to-Real Transfer**
>
> We thank the reviewer for this question. Sim-to-real transfer for soft robots in fluid is a big challenge for the field, requiring a multi-disciplinary effort that extends beyond the scope of a single work like ours. Bridging the sim-to-real gap is a well-known challenge for the field, involving significant hurdles in system identification, manufacturing inconsistencies, and sensor/actuator dynamics. Our work focuses on the critical prerequisite of establishing a robust simulation framework to discover optimal policies under idealized conditions, providing a vital baseline for future work dedicated to this challenge.
>
> ---
>
> **[W2] Lack of Joint Optimization**
>
> Thank you for your comment. Co-designing soft robots is an important emerging topic, yet it faces challenges in scaling to high-fidelity 3D simulations and in developing unified design representations. While existing works have made valuable progress, they often operate within certain constraints, such as simplified 2D design space (e.g., EvoGym [3]) or challenges in efficiently discovering controllers for complex 3D environments (e.g., SoftZoo [4]). Although our work focuses on controller optimization only, the framework we have developed shows a promising sign of tackling both of these challenges, providing a unified control representation within a high-fidelity 3D environment. We will tone down the co-design claim in our manuscript and revise its text accordingly to better clarify the position of our work.
>
> ---
>
> **[W3, Q1] General Controllers across Unseen Shapes**
>
> We thank the reviewer for raising this important point. While significant progress has been made in learning a Universal Controller for different rigid robot shapes in terrestrial locomotion tasks (e.g., MetaMorph [5]), transferring this capability to soft-bodied agents in fluid environments introduces substantial new difficulties due to the infinite-dimensional nature of soft-body deformations and complex fluid interactions.
>
> Tackling this infinite-dimensional challenge requires a **compact, low-dimensional representation** that can jointly encode a soft agent's morphology and its actuation across diverse robot shapes. We believe this is the key to solving the generalization problem. Our work offers a promising approach for the actuation component of this representation. The proposed automated, universal method for defining control points can be applied to general soft-body mesh. This could be combined with established techniques for morphology representation (e.g., graph neural networks or point clouds) to form a complete, universal input for a generalized control policy.
>
> Therefore, while our current work focuses on per-shape training, we see it as providing a promising component needed to develop universal controllers across various shapes in the future.
>
> ---
>
> **[W4]** Performance under Disturbance
>
> Thank you for suggesting this study. To address this, we have conducted new experiments to evaluate the robustness of our learned controllers.
>
> We tested robustness in a variant of the 2D trajectory tracking task (see Section 4.3 in supplementary materials for details): a straight-line moving target point guided the fish to a fixed goal while random lateral flow disturbances (width ≈ 0.3 body lengths) were applied. We define the following success rate to measure the swimmer's performance under disturbance and higher success rate indicates the policy is more capable of resisting disturbance: a trial is considered successful if the agent manages to reach the target point (defined as being within a distance of 0.2 units) within the episode's time limit after being disturbed. We conducted **50 independent trials** with randomized disturbance positions to ensure a robust evaluation. Across 7 disturbance velocities  reported in the table below, success rates remained high at 96% for disturbance velocity as large as 0.067 (c.f. the fish's swimming speed ≈ 0.05) and 68% at 0.1. These results indicate our trained controller's robustness under moderate disturbance.
>
> |Disturbance Velocity| 0.0 |0.033|0.067|0.1|0.133|0.167|0.2|
> |:---:|:---:|:---:|:---:|:---:|:---:|:---:|:---:|
> |Success Rate|100% |100%|96%|68%|42%|14%|4%|
>
> The current validation was performed on a 2D navigation policy trained in a quiescent fluid environment (see Section 4.3 in supplementary material). We expect that incorporating disturbances directly into the training curriculum, for instance through domain randomization, would likely lead to even greater policy robustness.
>
> ---
>
> **[W5 Q2] Computational Cost and Training Speed**
>
> Our simulator achieves frame rates of approximately 70 FPS for 2D morphologies and 30 FPS for 3D morphologies. Given our specified time step and episode duration (detailed in Supplementary Section 3), a single episode completes in approximately 85 seconds for 2D cases and 100 seconds for 3D cases. For comparison, at an equivalent fluid resolution, the DiffMPM-based simulator used in SoftZoo[4], one of the SOTA works on soft swimmers, requires approximately 350 seconds to complete a forward simulation of the same duration for its 3D scenes.
> As for the training time, a complete training run for one agent, consisting of 75000 environment steps (equivalent to 3 million simulation steps), typically takes 3-4 hours with parallel environments on 8 GPUs.
>
> ---
>
> **[Q3] Reward Function and Weights**
>
> Each reward functions has a clear physical motivation:
> - **The smoothness penalty** is the difference of action vectors between adjacent frames, encouraging more continuous actions, a common technique in robotics to reduce jittery motions.
> - **The action penalty** is the norm of the action vector that prevents excessive deformation.
>
> We will add more details on them to our supplementary material.
>
> To determine proper reward weights, we began by tuning the primary task reward to achieve basic locomotion, then incrementally adjusted the regularization weights to refine the emergent gaits—for instance, increasing the smoothness penalty to mitigate jittery movements. This process led us to a set of baseline weights (w_s = -0.02, w_a = -0.01) that produced stable and effective gaits, which we then used consistently across all experiments for fair comparison.
>
> ---
>
> **[Q3] Poor Tuned Reward Weights**
>
> Yes, a poorly tuned reward can limit the emergent strategy. **Extreme reward weights lead to degenerated, albeit explainable solutions**. An excessively high action penalty, for example, teaches the agent to remain nearly stationary, a trivial locomotion strategy that correctly minimizes the action penalty. Such cases are usually straightforward to identify and fix during initial tuning and confirm that the reward terms function as intended.
>
> We are truly grateful for your insightful comments and constructive feedback. We hope that the additional details and experimental results we provided have effectively addressed your concerns. Please feel free to let us know if there are any remaining concerns. We look forward to hearing your thoughts on our rebuttal.
>
> ---
>
> References:
>
> [1]. Triantafyllou, M. S., Triantafyllou, G. S., & Yue, D. K. (2000). Hydrodynamics of fishlike swimming. Annual review of fluid mechanics, 32(1), 33-53.
>
> [2]. Lin, Z., Hess, A., Yu, Z., Cai, S., & Gao, T. (2019). A fluid–structure interaction study of soft robotic swimmer using a fictitious domain/active-strain method. Journal of Computational Physics, 376, 1138-1155.
>
> [3]. Bhatia, J., Jackson, H., Tian, Y., Xu, J., & Matusik, W. (2021). Evolution gym: A large-scale benchmark for evolving soft robots. Advances in Neural Information Processing Systems, 34, 2201-2214.
>
> [4]. Wang, T. H., Ma, P., Spielberg, A. E., Xian, Z., Zhang, H., Tenenbaum, J. B., ... & Gan, C. (2023). SoftZoo: A Soft Robot Co-design Benchmark For Locomotion In Diverse Environments. In The Eleventh International Conference on Learning Representations.
>
> [5]. Gupta, A., Fan, L., Ganguli, S., & Fei-Fei, L. (2022). MetaMorph: Learning Universal Controllers with Transformers. International Conference on Learning Representations.

---

> > ### Author Response · Authors · 2025-08-06
> >
> > Dear Reviewer,
> >
> > Thank you again for your review. We wanted to follow up on our rebuttal to see if you have any remaining questions. We would be happy to provide any further clarifications.
> >
> > Thank you for your time and consideration.

---

> > > ### Comment · Reviewer_xudV · 2025-08-06
> > >
> > > Thanks for the rebuttal and the clarifications.

---

> ### Comment · Area_Chair_6hCr · 2025-08-06
> **reviewer response needed**
>
> Hello,
>
> The last step in the reviewing process is to process the updates from the authors that are key in clearing up final issues to ensure papers get a fair treatment. Please respond to the reviewers ASAP to address any final threads.
>
> - AC

---

### Official Review · Reviewer_Z1aA · 2025-07-03

**Clarity:** 3
**Significance:** 3
**Originality:** 3
**Rating:** 4
**Confidence:** 3

**Summary:**

This paper proposes a novel, automated framework for learning control policies for soft-bodied swimmers with arbitrary morphologies. It introduces a unified, reduced-mode control space using geodesic sampling and linear blend skinning to enable morphology-agnostic actuation. A GPU-accelerated simulator based on the Lattice Boltzmann Method is developed to efficiently capture fluid-structure interactions at high fidelity. The method is evaluated across a diverse set of morphologies, including biologically inspired and abstract forms, and demonstrates improved performance over state-of-the-art (SOTA) baselines in terms of swimming effectiveness.

**Questions:**

- You state that your method discovers swimming strategies “similar to those found in nature.” How do you measure this similarity? Have you considered comparing swimming trajectories or gait metrics with biological data?

- Does your method account for or optimize the energetic efficiency of swimming? If not, could your framework be extended to incorporate energy-based objectives ?

- Have you considered deploying these controllers on physical soft robots? What challenges do you anticipate in transferring them from simulation to the real world?

**Ethical Concerns:**

["NO or VERY MINOR ethics concerns only"]

**Final Justification:**

The authors addressed my concerns and I will keep my score of 4 (boderline accept). For a full accept and increased impact, it would have needed to included the co-design of bodies and morphologies.

**Limitations:**

yes.

**Quality:**

3

**Strengths And Weaknesses:**

In general, I found the paper well written and an interesting contribution. In particular:

Strengths:
- The proposed reduced-mode controller enables efficient and expressive actuation across diverse geometries, reducing the need for expert-designed actuators.
- The use of LBM for fluid dynamics and a custom GPU-accelerated simulator ensures physically realistic interactions critical for learning effective swimming strategies. The simulator is a good contribution to the research community.
- The method generalizes to unconventional morphologies where baseline methods fail, showing the strength of a data-driven, morphology-agnostic approach.

Weaknesses:
- The framework does not seem to optimize or evaluate energy usage. While action magnitudes are regularized, there is no explicit metric  to assess or minimize energy consumption.
- Claims about the similarity between learned behaviors and evolved biological swimming strategies are based on qualitative visual resemblance only. No quantitative biomechanical comparisons or gait analyses are provided.
- Despite referencing "co-design," the work only addresses control learning on fixed shapes, and does not co-optimize morphology and control jointly.

In general, especially the abstract oversells the results in my opinion. For example: "We showcase how our automated approach discovers high-performance swimming patterns similar to those found in nature over years of evolution." Without a more detailed analysis, this claim does not seem to substantiated. Additionally, energy efficiency is a crucial aspect of biological organisms, and doesn't seem considered here.

Additionally,  the authors write that"Our work unlocks future opportunities in automated co-design of morphology and control of soft robots in complex hydrodynamic environments." Why does this work in particular unlock this possibility and not existing works on soft robots?

---

> ### Author Rebuttal · Authors · 2025-07-31
>
> We sincerely thank the reviewer for the insightful and constructive feedback on our manuscript. We found the comments regarding energy efficiency, quantitative biological comparison, the scope of co-design, and sim-to-real transfer to be particularly valuable. These suggestions have helped us to significantly strengthen the evaluation and framing of our work.
>
> > **[W1]** The framework does not seem to optimize or evaluate energy usage. While action magnitudes are regularized, there is no explicit metric to assess or minimize energy consumption. **[Q2]** Does your method account for or optimize the energetic efficiency of swimming? If not, could your framework be extended to incorporate energy-based objectives ?
>
> We thank the reviewer for raising this point. Energy efficiency is a key performance metric for swimming, and we prepared **a new experiment** to optimize the energetic efficiency of soft swimmers in our framework. We follow established methodologies from the biomechanics literature [1] and adopt a physically-grounded measure of energy consumption. Our framework is designed such that this **energy cost** (see table below) can be directly and efficiently calculated at each simulation step. This allows us to incorporate it as a penalty term within the reward function, enabling the training of policies that explicitly optimize for energy efficiency.
>
> To demonstrate that our framework is capable of learning **energy-efficient gaits**, we have trained policies with different penalty coefficients (w_e) for this energy cost on the 3D clownfish model, which we selected as a representative case study. We believe the findings from this model are indicative of the framework's general capabilities across all our tested morphologies. The results from this experiment are summarized below:
>
> |w_e|Forward Distance|Energy Cost|Cost of Transport (CoT)
> |---|---|---|---|
> 0 |1.68 m |433.2 J |258.3 J/m
> 0.005| 1.57 m |211.0 J| 134.3 J/m
> 0.02 |1.06 m| 78.8 J | **74.7 J/m (Optimal)**
> 0.05| -0.01 m |0.3 J|-
>
> To analyze gait efficiency, we adopted the widely-accepted Cost of Transport (CoT) metric, which normalizes energy consumption by distance traveled (lower is better). Our results demonstrate a clear trade-off between distance and efficiency. As the penalty increases ($0\to 0.005\to 0.02$), the trained policy demonstrates better **energetic efficiency of swimming**, while an excessive penalty (0.05) correctly produces a near-stationary policy. This validates that our framework robustly optimizes for energy-based objectives in a logical manner, and we believe these results fully address the reviewer's concerns.
>
> ---
>
> > **[W2]** Claims about the similarity between learned behaviors and evolved biological swimming strategies are based on qualitative visual resemblance only. No quantitative biomechanical comparisons or gait analyses are provided. **[Q1]** You state that your method discovers swimming strategies “similar to those found in nature.” How do you measure this similarity? Have you considered comparing swimming trajectories or gait metrics with biological data?
>
> Thank you for your question.
>
> Our comparison to biological swimming is indeed primarily qualitative. A direct quantitative analysis presents non-trivial challenges, particularly in acquiring precise biological kinematic data and in bridging the significant sim-to-real gap in matching animal physiology.
>
> We would like to add that our qualitative similarity extends beyond visual resemblance to a more scientific, **taxonomy correspondence**. Previous biomechanical research has identified several canonical swimming modes in aquatic animals, including **undulation** (eel), **pulsation** (jellyfish) and **oscillation** (paramecium) propulsion [2]. We have confirmed that our fluid-solid simulation can reproduce these three mechanisms.
>
> Our unified framework, without any expert knowledge as input, spontaneously exhibits multiple biological swimming modes across diverse swimmer morphologies during training. The emergence of such diverse behaviors has a biological root and is a non-trivial result, which we largely attribute to the physical fidelity of our simulation. In contrast, prior works in soft swimmer learning have often focused on or exhibited undulation swimming mode only (e.g., DiffAqua [3], which focuses on anguilliform motion). This demonstrates a significant degree of generality in our approach.
>
> We will carefully revise our wording in the manuscript to emphasize that our findings have a **qualitative connection** to these canonical biological swimming modes, and clarify our similarities with them to better position our work.
>
> ---
>
> > **[W3]** Despite referencing "co-design," the work only addresses control learning on fixed shapes, and does not co-optimize morphology and control jointly.
>
> Thank you for your comment. Co-designing soft robots is an important emerging topic, yet it faces challenges in scaling to high-fidelity 3D simulations and in developing unified design representations. While existing works have made valuable progress, they often operate within certain constraints, such as simplified 2D design space (e.g., EvoGym [4]) or challenges in efficiently discovering controllers for complex 3D environments (e.g., SoftZoo [5]). Although our work focuses on controller optimization only, the framework we have developed shows a promising sign of tackling both of these challenges, providing a unified control representation within a high-fidelity 3D environment. We will tone down the co-design claim in our manuscript and revise its text accordingly to better clarify the position of our work.
>
> ---
>
> > **[Q3]** Have you considered deploying these controllers on physical soft robots? What challenges do you anticipate in transferring them from simulation to the real world?
>
> We thank the reviewer for this question. Sim-to-real transfer for soft robots in fluid is a big challenge for the field, requiring a multi-disciplinary effort that extends beyond the scope of a single work like ours. The path to physical deployment of soft robotic swimmers involves at least three significant hurdles, which are shared challenges across the soft robotics community. First, **modeling** requires non-trivial and probably labor-intensive system identification of physical parameters like fluid viscosity, material stiffness and damping, to narrow the sim-to-real gap. Next, **manufacturing** introduces issues of consistency and fabrication imperfections, meaning any physical prototype will have variations not present in the idealized digital model. Finally, **deployment** must contend with real-world complexities such as sensor noise, state estimation errors, and actuation latency. Addressing these challenges constitutes a significant research topic in its own right. Our current work focuses on a critical prerequisite: establishing a robust framework to discover what an optimal policy looks like under idealized conditions. We believe this provides a vital baseline and a clear target for future research dedicated to addressing the well-established challenges of sim-to-real transfer.
>
> We are truly grateful for your insightful comments and constructive feedback. We hope that the additional details and experimental results we provided have effectively addressed your concerns. Please feel free to let us know if there are any remaining concerns. We look forward to hearing your thoughts on our rebuttal.
>
> ---
>
> References:
>
> [1]. Verma, S., Novati, G., & Koumoutsakos, P. (2018). Efficient collective swimming by harnessing vortices through deep reinforcement learning. Proceedings of the National Academy of Sciences, 115(23), 5849-5854.
>
> [2]. Zhang, D., Zhang, J. D., & Huang, W. X. (2022). Physical models and vortex dynamics of swimming and flying: A review. Acta Mechanica, 233(4), 1249-1288.
>
> [3]. Ma, P., Du, T., Zhang, J. Z., Wu, K., Spielberg, A., Katzschmann, R. K., & Matusik, W. (2021). Diffaqua: A differentiable computational design pipeline for soft underwater swimmers with shape interpolation. ACM Transactions on Graphics (TOG), 40(4), 1-14.
>
> [4]. Bhatia, J., Jackson, H., Tian, Y., Xu, J., & Matusik, W. (2021). Evolution gym: A large-scale benchmark for evolving soft robots. Advances in Neural Information Processing Systems, 34, 2201-2214.
>
> [5]. Wang, T. H., Ma, P., Spielberg, A. E., Xian, Z., Zhang, H., Tenenbaum, J. B., ... & Gan, C. SoftZoo: A Soft Robot Co-design Benchmark For Locomotion In Diverse Environments. In The Eleventh International Conference on Learning Representations.

---

> > ### Comment · Reviewer_Z1aA · 2025-08-04
> >
> > Thank you for the clarification; that was very helpful. I'll follow the discussion with the other reviewers before revising my score.

---

> > > ### Author Response · Authors · 2025-08-05
> > >
> > > Thank you very much for your feedback. We are pleased to hear that our clarification was helpful.
> > >
> > > We are happy to provide any further clarification that might be needed during your discussion.

---

### Official Review · Reviewer_NiLK · 2025-07-04

**Clarity:** 2
**Significance:** 3
**Originality:** 3
**Rating:** 4
**Confidence:** 2

**Summary:**

This paper introduces a new framework for learning control policies for soft-bodied swimming robots with diverse shapes. Unlike traditional approaches that rely on human-designed muscle patterns or simplified fluid simulations, this method uses a low-dimensional, morphology-agnostic control space inspired by how real animals swim. The authors also develop a fast, GPU-based simulator that accurately models the interaction between the swimmer and the surrounding fluid, which is essential for learning realistic swimming strategies.

The system automatically generates controllers by sampling over reduced deformation modes, removing the need for expert-designed heuristics. Experiments show that the method achieves significantly better performance than existing baselines—especially on unusual shapes like toruses—and can learn advanced behaviors such as exploiting fluid vortices. The results suggest that this framework could support future work in co-designing both the shape and control of soft robots in complex environments.

**Questions:**

1. What is the simulation frame rate (FPS) achieved for the soft-body fluid-structure simulation?

2. How many training runs were conducted, and how consistent is the performance across different seeds and across the training.

**Ethical Concerns:**

["NO or VERY MINOR ethics concerns only"]

**Final Justification:**

I've read the rebuttal by the authors, and my concerns are reasonably addressed. In this case, I'm keeping my original evaluation as borderline accept.

**Limitations:**

Yes.

**Quality:**

3

**Strengths And Weaknesses:**

# Strengths

1. The paper presents a well-integrated framework for soft-body swimming control that combines a GPU-accelerated simulator, a two-way fluid-structure interaction model, and a unified control representation for diverse morphologies. This integration addresses bottlenecks in simulating and learning soft-body locomotion, offering both physical fidelity and computational efficiency.

2. The proposed method demonstrates robust performance not only on biologically-inspired morphologies but also on abstract and unconventional shapes. This highlights the generality and adaptability of the control approach, which does not rely on human-designed heuristics and can discover effective strategies in previously underexplored morphospace.

3. The paper provides high-quality visualizations of the deformation modes, control points, and learned swimming trajectories, which help clarify the system’s behavior and effectiveness. These visual results significantly aid in understanding the qualitative performance of the method.

# Weaknesses:

1. The core contributions primarily lie in the modeling of the soft-body and its interaction with fluid dynamics, while the learning approach itself follows established reinforcement learning method. As such, the work may be better suited to other venues

2. The evaluation largely relies on task-specific rewards that may include regularization or smoothing terms, which can obscure direct assessment of locomotion quality. Inclusion of more interpretable and standardized metrics—such as forward swimming speed, displacement over time, or time-to-target—would provide clearer insight into the quantitative effectiveness of the learned policies.

3. Given the stochastic nature of reinforcement learning, it is important to report training consistency and stability. However, the manuscript does not specify how many training trials were repeated or whether the final performance is robust across seeds. Including training curves and variance across runs would strengthen the empirical evaluation.

---

> ### Author Rebuttal · Authors · 2025-07-31
>
> We sincerely thank you for your recognition of our learning framework and visualizations, as well as your constructive feedback regarding the evaluation metrics and training stability analysis. We address each of the weaknesses and questions raised below.
>
> > **[W1]** The core contributions primarily lie in the modeling of the soft-body and its interaction with fluid dynamics, while the learning approach itself follows established reinforcement learning method. As such, the work may be better suited to other venues.
>
> We thank the reviewer for recognizing our contributions in modeling the soft-body and its interaction with fluid dynamics.
>
> Our work directly addresses the emerging topic of soft robot control in complex physical environments, a subject of growing interest within the learning community (e.g., recent works at NeurIPS [1,2] and ICLR [3]).
>
> While our work builds upon established RL algorithms, we believe our contributions are of direct service to the learning community in the following ways:
>
> - **A unified and effective design** for low-dimensional control of general soft objects. This enables the learning community to scale up prior research from simple morphologies to a much broader and more expressive class of soft robots.
> - **A high-fidelity solid-fluid simulator**. This provides the community with a challenging new benchmark for developing and testing RL algorithms on more physically realistic swimming behaviors.
> - **Empirical validation** of RL's feasibility for diverse soft swimmers. This result, demonstrating that RL can learn effective gaits for non-traditional soft objects, is a novel and non-trivial finding in itself, opening up new research questions for the community.
>
> In summary, we believe our work is a strong fit for the NeurIPS community, and we hope these contributions will foster further research into this challenging and important domain.
>
> ---
>
> > **[W2]** The evaluation largely relies on task-specific rewards that may include regularization or smoothing terms, which can obscure direct assessment of locomotion quality. Inclusion of more interpretable and standardized metrics—such as forward swimming speed, displacement over time, or time-to-target—would provide clearer insight into the quantitative effectiveness of the learned policies.
>
> Thank you for the question. We would like to clarify that the normalized reward reported in the tables in our manuscript has excluded all regularization terms, purely reflecting the forward swimming distance achieved within the time limit. For example, the data in Table 1(Section 6.2, Quantitative Results), when divided by 5 (action signal frequency in our experiment), gives the actual movement distance (in meters). Regarding the forward swimming speed metric, as the forward swimming task fixes the time horizon, the speed is in direct proportion to the reported swimming distance (i.e., dividing it by the fixed time horizon). From table1, It can be observed that our optimized swimming strategies have an average speed ranging from 0.1 to 0.3 m/s (the body length of the swimmers is around 1 meter).
>
> For the turning task in our supplementary material, as well as for the two new tasks we added in the rebuttal (energy-efficient swimming and station-keeping), we will add the standardized performance metrics you suggested to the revised manuscript.
>
> ---
>
> > **[W3]** Given the stochastic nature of reinforcement learning, it is important to report training consistency and stability. However, the manuscript does not specify how many training trials were repeated or whether the final performance is robust across seeds. Including training curves and variance across runs would strengthen the empirical evaluation.
> **[Q2]** How many training runs were conducted, and how consistent is the performance across different seeds and across the training.
>
> We fully agree that training stability should be validated given RL's stochasticity. To address this, we conducted additional experiments on the 2D fish morphology. We trained forward swimming policies across 6 random seeds with identical hyperparameters.
>
> We analyzed the intermediate rewards collected at 5%, 15%, 25%, 50% and 100% of training episodes. The results are reported below in the table.
>
> |Training Progress|Mean Smoothed Reward|Std. Dev.|
> |---|---|---|
> 5%|-0.0047          |0.0742
> 15%|4.6269|0.6278
> 25%|6.1198|0.9111
> 50%|6.7665|0.8122
> 100%|6.7135|0.7733
>
> From this data, we draw three key conclusions:
>
> - **Consistent Final Performance**: The low standard deviation in the final (100%) reward demonstrates that all runs reliably converge to policies of similar high quality (as a reference, the final average speed is about 0.09 m/s, while the standard deviation is only 0.01m/s).
> - **Stable Learning Trajectory**: The consistently low standard deviation at every milestone (25%, 50%, 75%) confirms the learning process itself is stable and reproducible.
> - **Effective Progress**: The monotonic increase in the mean reward across all milestones shows this stable process is also effective at making consistent progress.
>
> Collectively, these quantitative results demonstrate that our training framework is robust, and its performance is consistent and reproducible across different random seeds. We would be happy to provide more evaluations if requested.
>
> ---
>
> > **[Q1]** What is the simulation frame rate (FPS) achieved for the soft-body fluid-structure simulation?
>
> Thank you for your question regarding the computational cost.
>
> Our system achieves approximately 30 FPS for the 3D scenes presented in the paper, using a single GPU. To provide a concrete example, the clownfish in the forward swimming task (see Fig. 2), which uses a 128x128x512 fluid grid, a fish model comprising 3,543 finite elements, and a time step of 0.005s, runs at an average of 29.9 FPS.
>
> Further details on the simulation parameters can be found in Section 3 of the supplementary material.
>
> We are truly grateful for your insightful comments and constructive feedback. We hope that the additional details and experimental results we provided have effectively addressed your concerns. Please feel free to let us know if there are any remaining concerns. We look forward to hearing your thoughts on our rebuttal.
>
> ---
>
> References:
>
> [1] Bhatia, J., Jackson, H., Tian, Y., Xu, J., & Matusik, W. (2021). Evolution gym: A large-scale benchmark for evolving soft robots. Advances in Neural Information Processing Systems, 34, 2201-2214.
>
> [2] Wang, T. H. J., Zheng, J., Ma, P., Du, Y., Kim, B., Spielberg, A., ... & Rus, D. (2023). Diffusebot: Breeding soft robots with physics-augmented generative diffusion models. Advances in Neural Information Processing Systems, 36, 44398-44423.
>
> [3] Wang, T. H., Ma, P., Spielberg, A. E., Xian, Z., Zhang, H., Tenenbaum, J. B., ... & Gan, C. SoftZoo: A Soft Robot Co-design Benchmark For Locomotion In Diverse Environments. In The Eleventh International Conference on Learning Representations.

---

> > ### Author Response · Authors · 2025-08-06
> >
> > Dear Reviewer,
> >
> > Thank you again for your review. We wanted to follow up on our rebuttal to see if you have any remaining questions. We would be happy to provide any further clarifications.
> >
> > Thank you for your time and consideration.

---

> ### Comment · Area_Chair_6hCr · 2025-08-06
> **Reviewer response needed**
>
> Hello,
>
> The last step in the reviewing process is to process the updates from the authors that are key in clearing up final issues to ensure papers get a fair treatment. Please respond to the reviewers ASAP to address any final threads.
>
> - AC

---

### Note · Authors · 2025-08-12

Dear AC and Reviewers,

We are grateful for the constructive feedback and engaging discussion period. We would like to take this final opportunity to summarize our contributions and highlight how the review process has strengthened the paper.

The control of soft robots in complex physical environments is a key challenge with great potential for robotics and AI. Progress in this area has often been limited by simplified physics or geometries. Our work contributes a high-fidelity simulation platform paired with a unified, low-dimensional control representation to directly address these limitations. We are confident that our work will encourage more exploration of learning in high-fidelity physical simulations and the control of more expressive soft robot morphologies within the community.

We are sincerely grateful to the reviewers for their rigorous feedback, which has been instrumental in improving this manuscript. Specifically, prompted by their suggestions, we demonstrated greater behavioral diversity through new tasks like energy-efficient locomotion and flow resistance. We also confirmed our method's robustness with a comprehensive stability analysis across multiple seeds. Finally, we performed new experiments showing the learned controllers can handle external disturbances, confirming they are robust, closed-loop policies.

We believe these revisions have addressed the key questions raised and confirmed the soundness of our work. Thank you again for your time and valuable feedback.

---

### Decision · Program_Chairs · 2025-09-17

**Decision:**

Accept (poster)

**Comment:**

Most reviewers agree that the paper should be accepted.

The strengths of designing a new approximation for soft body control and faster GPU-accelerated outweigh the limited morphology diversity.